# Prevalence and factors associated with experience of corporal punishment in public schools in South Africa

**Pinky Mahlangu**[1,2]*, **Esnat Chirwa**[1,2], **Mercilene Machisa**[1,2], **Yandisa Sikweyiya**[1,2], **Nwabisa Shai**[1,2], **Rachel Jewkes**[1,2]

**1** Gender & Health Research Unit, South African Medical Research Council, Pretoria, South Africa, **2** School of Public Health, University of the Witwatersrand, Johannesburg, South Africa

* pinky.mahlangu@mrc.ac.za

**Data Availability Statement:** All data used in the analysis of the manuscript are available on the SAMRC data repository: http://medat.samrc.ac.za/index.php/catalog/45.

## Abstract

### Background

Corporal punishment (CP) is still a common practice in schools globally. Although illegal, studies in South Africa report its continued use, but only a few have explored factors associated with school CP. Moreover, extant studies have not shown the interrelationships between explanatory factors. This study aimed to determine the prevalence and factors associated with learners' experiences, and to examine pathways to the learners' experiences of CP at school.

### Method

3743 grade 8 learners (2118 girls and 1625 boys) from 24 selected public schools in Tshwane, South Africa, enrolled in a cluster randomised controlled trial evaluating a multi-component school-based intervention to prevent intimate partner violence, and completed self-administered questionnaires. We carried out descriptive analysis, simple linear and structural equation modelling to examine factors and pathways to the learners' experience of CP at school.

### Results

About 52% of learners had experienced CP at school in the last 6 months. It was higher among boys compared to girls. Experience of CP at school amongst learners was associated with learner behavior, home environment, and school environment. Learners from households with low-socio economic status (SES) had an increased risk of CP experience at school. Amongst boys, low family SES status was associated with a negative home environment and had a direct negative impact on a learner's mental health, directly associated with misbehavior.

### Conclusion

CP in public schools in South Africa continues despite legislation prohibiting its use. While addressing learner behaviour is critical, evidence-based interventions addressing home and

**Funding:** RJ received funding from an anonymous donor who had no role in the study; RJ received funding from the South African Medical Research Council. These sponsors played no role in the research. No research costs or authors' salaries were funded, in whole or in part, by a tobacco company. The authors are not aware of any competing interest from the donor. We do not think that the identity of the donor might be considered relevant to editors or reviewers' assessment of the validity of the work. The donor had no involvement in the writing of the manuscript other than funding the study. The authors are not aware of any competing interests.

**Competing interests:** The authors are not aware of any competing interests.

school environment are needed to change the culture among teachers of using corporal punishment to discipline adolescents and inculcate one that promotes positive discipline.

## Introduction

While the use of a physical methods of discipline, also referred to as corporal punishment (CP) at school is banned and no longer an issue in European countries; it continues to be prevalent in a third of the world's countries, despite evidence regarding its harmful physical, mental and behavioural effects on the child [1, 2]. The global prevalence of CP in schools ranges between 13%– 97% of learners who reported experience of CP at school [1]. Learners continue to experience CP in most countries in Sub-Saharan Africa despite legislation prohibiting its use [1, 2]. The South African Schools Act No 84 of 1996 states that no person may administer CP to a learner at school [3]. Any person who administers corporal punishment at school is guilty of an offence and liable on conviction to a sentence which could be imposed for assault [3]. Corporal punishment has been an integral part of schooling for most teachers and learners in twentieth century South African schools, characterized by a legacy of authoritarian education practices under Bantu education, and a belief that CP is necessary for orderly education [4]. The ending of apartheid and the establishment of a human rights culture in the 1990s laid the foundation for legislation aimed at ending use of CP in schools in South Africa [4].

South Africa is a highly inequitable society and this inequality is reflected in the historically complex two-tier education system namely, the middle class, private schools and the public schools [5]. The middle class, formerly white schools no longer use corporal punishment as a discipline method [4]. However, in public schools, use of corporal punishment is still common practice [6, 7]. In most public schools, classrooms are overcrowded and under-resourced, and teachers are often under-qualified and overworked [8, 9]. Teachers feel disempowered and ill-equipped with viable alternative discipline methods to maintain a safe and secure environment which facilitates learning [6]. A national study conducted in 2012 showed that 49.8% of the 5939 learners had been caned or spanked by a teacher or principal in South African schools [10]. Use of CP in public schools in South Africa is a nation-wide problem that warrants further targeted research to inform prevention and responses [10].

Studies have been conducted to understand the reasons underlying the continued use of CP in schools in South Africa. CP is administered for both serious and minor offences such as being absent from school or not doing homework, not knowing the answer to the questions asked by teachers in class, caring a gun at school, and for talking or disrupting a lesson in class [6, 11]. Many teachers believe that CP is an effective method of correcting deviant behaviour and maintaining discipline in the classroom [12, 13].

While previous research has focused on determining the prevalence of learner experiences of CP in schools, limited research has focused on understanding what increases the risk of individual learners experiencing CP at school in South Africa. To contribute to this knowledge gap, we present analysis of quantitative baseline data collected from grade 8 learners who enrolled in a Skhokho Supporting Success randomised controlled trial (RCT) [14, 15]. The analysis aimed to determine the prevalence of CP experienced by learners in selected public schools in the last 6 months, and to examine factors associated with their experiences of CP at school. Using structural equation modelling, we elucidated pathways to the learners' experience of CP at school. Understanding risk factors for learners' experience of CP at school may provide much needed evidence to inform interventions to curb the continued use of CP by teachers.

## Factors associated with experience of corporal punishment at school

Evidence from South Africa and elsewhere suggests that school CP is undergirded by a myriad of individual, school, family and broader community level risk factors [10, 16]. In a study conducted in Khayelitsha, a peri urban township in the Western Cape province in South Africa found that the transgressions that led to CP at school included not doing school or home work, coming late from break, not listening to teachers, giving wrong answers in class and making noise [7]. Boys were more likely to experience CP both at school and at home [16]. Learners who perform poorly at school are likely to be beaten by their teachers and by parents and caregivers at home with the aim to encourage improved academic performance [17]. Other studies suggest that children subjected to CP may engage in more aggression and misbehavior than those who are not [18, 19]. A large body of research on CP is on learners, and addresses the question of what makes them vulnerable, yet behaviour of teachers and what makes some teachers to use CP is also important. Research has shown that teachers who are overwhelmed with personal problems and believed that CP was effective in managing behavior in the classroom were likely to use CP against learners at school [20]. Contextual factors including low socio-economic class, ethnicity and race of learners were associated with experience of CP in public schools in the United States of America [21].

Family background, in particular, low socio-economic status is associated with experience of CP at school [16]. Children who experience financial lack, who are exposed to violence, and those who do not receive affection and love from home are likely to misbehave and to experience CP at school [20]. Food insecurity is also a risk factor directly associated with the experience of school CP amongst boys and girls [16]. Buller, Hidrobo [22] found that food insecurity and violence exposure is linked to increased levels of household stress over the lack of resources, which further leads to the use of corporal punishment.

There are broader community and societal influences to children's experiences of CP. In communities where use of harsh discipline strategies is acceptable and where CP is perceived as effective in controlling behaviour, learners tend to be at increased risk of experiencing CP [23, 24]. Learners experiences of CP is also higher when the use of corporal punishment is justified to maintain discipline and to enable academic success [25, 26]. Related to this belief, is a positive correlation between one's experience of CP during childhood, and the approval of its use as an adult [7, 25]. Childhood experience of CP legitimates violence by the stronger against the weaker and increases the chance of the child becoming violent [10, 27]. Exposure to violence and traumas in childhood which involves physical punishment has been found to increase the likelihood of being victimised or becoming a perpetrator in adulthood [28, 29]. While use of CP is illegal in South Africa, there has been limited concerted effort to enforce the law, ensuring that those who continue to use CP are convicted of an offense; and training of teachers on alternative methods of classroom management and discipline has been inadequate [30, 31]. Given this background, it is important that we understand why some children are more likely to experience CP than others.

## Methods

### Study design

The Skhokho Supporting Success primary prevention study was a cluster randomized controlled trial (RCT) that sought to develop and evaluate a multi-faceted school-based interventions to prevent intimate partner violence [14, 15]. The evaluation sought to show a reduced IPV incidence among learners in the intervention arm. The trial was conducted in 2015 and 2016 among secondary public schools located in Tshwane District, Gauteng Province, South

Africa [14, 15]. Schools were randomized into three arms. The first arm received school strengthening including support for life orientation teaching (life skills, and guidance and counselling) and parent teenager relationship intervention; the second arm comprised of the school strengthening intervention only (including support for life orientation teaching); and the control arm received no intervention. The school strengthening intervention involved strengthening the institutional capacity of the school in teaching and practice of positive discipline, life skills and human rights and responsibilities [14, 15]. There were no significant differences in the number of learners and teachers in class.

Schools were selected if they were enrolling Grade 8 learners in 2014. All Grade 8 learners in 24 purposively selected English medium State secondary schools within a 50km radius from Pretoria City were eligible to participate in the study. While most (21 of 24) schools were located in black townships including Mabopane, Soshanguve, Hammanskraal, and Mamelodi, Winterveld and Atteridgeville, two were located in the central business district of Pretoria and one in Laudium, a predominantly Indian community [15]. Participants were recruited between February and April 2014.

## Ethical considerations

The South African Medical Research Council Ethics Committee gave ethics approval for the trial. Permission to work in schools was given by the Gauteng Department of Basic Education (DBE), at the provincial and district level. Permission was also granted by the school principals in the 24 schools. The information letter detailing the nature and purpose of the study, the interventions, participants, what the project would offer to schools, risks and benefits of participating in the study, project timelines, and rights of participants was provided to principals. School principals were also informed that all data generated from the study will be kept confidential and that research reports and articles which will be submitted in scientific journals will not include any information that may identify the school or any of the educators, learners, school governing body officials and parents. All Grade 8s from the 24 schools were eligible to participate in the trial. Participants (learners) gave written assent and consent was obtained from their parents or caregivers before they were enrolled in the trial. All learners in Grade 8 in the participating schools were given a pocket booklet with phone numbers of local professional services which offer help on psychosocial, violence and substance abuse and other challenges. Any learner who specifically identified him or herself as in need of help during the study period was, with her consent, taken by teachers to a relevant local service, as per DBE standard practice. There was no financial re-imbursement for participating in the trial. The trial is registered on ClinicalTrials.gov as NCT02349321. The combined total school enrolment was 6076 of which 4095 (67.4%) obtained consent from the parents or caregivers to participate. Of the 4095, 3811 (93.1%) provided assent and participated in the trial. The percentage of learners in each school that did not complete the survey (incomplete surveys) ranged from 0.6% to 7.0% (mean 3.8%, SD 1.81).

## Data collection

A survey was conducted using a self-administered questionnaire loaded in personal digital assistants (PDAs). The questionnaires were presented in English and two local languages spoken in the study area, SeTswana and SePedi. A participant could choose to use any of the three languages, either through text or voice to complete the questionnaire. Participants privately completed the survey in class and could ask for help from trained fieldworkers who oversaw the data collection process. After the survey, the data were uploaded to a secure server only accessible to the investigators and downloaded for cleaning and analysis [14, 15].

**Table 1. Description of the factors that influence experience of corporal punishment at school.**

| Latent factor | Measures | Description of variables |
|---|---|---|
| Home Environment | Male and female care-giver kindness/ support | An additive score from 7 items, measured on a 5-point likert scale (strongly agree (1), agree (2), disagree (3), strongly disagree (4), no caregiver (5)). Example item "My mother/ female caregiver does everything she can to support me." |
| | Caregiver-learner communication. | An additive score from 4 items, measured on a 5-point scale (everyday, each week, at least once a month, sometimes but not each month, never). Example item "How often does one of your parents or caregivers ask how you are feeling or whether anything bothers you?" |
| | Learner's experience of physical punishment. | An additive score of 3 items derived from the Child Trauma Questionnaire [32]. Example item "I have been punished at home by being beaten every day or every week" with responses as—in past 6m, between 6m and 12m, before the last 12m, never. |
| | Learner's experience of neglect in the home. | An additive score of 2 items derived from the Child Trauma Questionnaire [32]. Example item "I have spent time outside the home and none of the adults at home knew where I was", with responses as—in past 6m, between 6m and 12m, before the last 12m, never. |
| Learner's behaviour | Learner's substance use (alcohol or drugs) | An additive score from 2 item on learner's alcohol use or drug use. Example item: "How often do you usually drink alcohol?", with responses as: every week, every month, less than once a month, less than once in a year, never. |
| | Learner's sexual behaviour | An additive score derived from question on whether learner has ever dated, is having sexual relationships, is involved in transactional sex and the number of sexual partners ever had. |
| | Misbehaviour | An additive score derived from 5 items measured on a 4-point scale. Example item: "How often have you been involved in a fight with knives?", with responses as: never, once, 2–3 times, more often. |
| School climate | Care-giver's attitude to learner's school work | One item measure on learner's perception of care-giver attitude to schoolwork (My parents or care givers do not care how well I do at school), measure on a 4-point scale (Strongly agree-strongly disagree). |
| | Learner's perception of teachers' behaviour | An additive score of 7 items measuring the learner's perception of teachers' behavior, each item measured on a 4-point scale (strongly agree- strongly disagree). Example item: "Teachers are often late for class or miss lessons". |
| | Learner's attitude to other learners and teachers (bullying) | An additive score derived from 3 items on learner's bullying attitude, each item measured on a 4-point scale. Example item: "I like to make life difficult for our teachers by doing what I feel like no matter what they say" |

## Conceptual model and measures

This paper presents an analysis from the baseline data from the Skhokho. The binary outcome is the learner's experience of CP at school, derived from responses to one item: 'In the past 6 months were you ever beaten by a teacher?', with a 'yes or no' response.

Our hypothesized conceptual model looks at the inter-relationship between a learner's home environment, mental health, attitude to schoolwork, learner behavior, and experience of CP at school. These factors are derived from several measures as summarized in Table 1. Our conceptual model is based on the hypothesis that a learner's experience of CP at school is linked to their school environment (influenced by learner's attitude and behavior at school, their perception of their teacher's behavior and the attitude of their caregivers towards school work). The school environment is linked to home environment and mental health. The home environment is related to the family's socio-economic status, therefore we derived a socio-eco-nomic status score using the learners' parents/caregiver employment status, type of housing learner was living in, and amount of pocket money given to learner in a week (measured on an ordinal scale). We also hypothesized that learners' negative experiences at home (physical pun-ishment, neglect or poor communication with caregivers), their behavior (substance use, sex-ual behavior and misbehaviour, and their attitudes (towards schoolwork and towards teachers) were influenced by their age. Our hypothesized conceptual model is shown in Fig 1:

## Data analysis

We used baseline data from the trial, and we applied data analytical procedures which took into account the school clustered sampling design and the cross-sectional nature of the

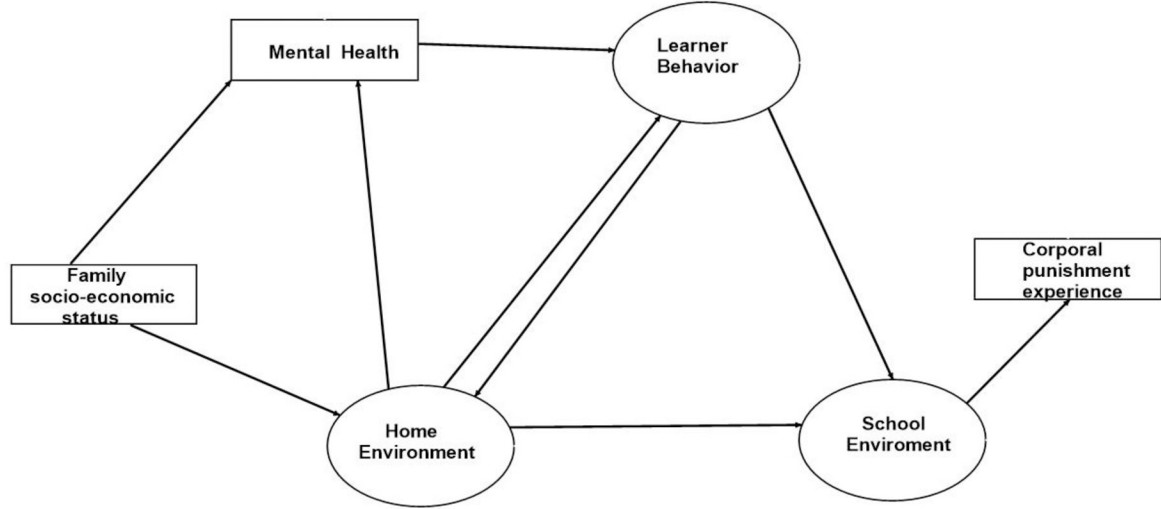

**Fig 1. Hypothesized conceptual model for experiencing corporal punishment at school.**

baseline survey. This was an analysis of the baseline data from a cluster RCT design, with participants clustered within schools. We carried out descriptive analysis on all potential explanatory factors associated with learner's experience of CP at school as described in our conceptual model. We used logistic regression to assess the relationship between experience of CP at school and all individual measures described in Table 1, and also assessed the correlation amongst the measures. We then assessed how the measures loaded unto our hypothesized latent constructs as defined in Table 1 and in Fig 1. We utilized various fit indices such as the (1) Comparative fit Index (CFI), (2) Tucker-Lewis Index (TLI), and Root Mean Square Error of Approximation (RMSEA), to assess how well the measurement model fitted to the observed data. For each factor, we allowed the variables to correlate freely and we also assessed the relationship between our outcome and each latent factor. All the 3 latent factors had high factor loadings and very good fit indices (CFI>0.95, TLI >0.95 and RMSEA<0.05). We then performed a Latent Path Analysis by applying general structural equation modelling techniques to assess the inter-relationships between the corporal punishment binary outcome, the latent constructs and other measured factors as hypothesized in our conceptual model.

The analyses were done separately for boys and girls and the final structural models are presented in Figs 2 & 3. The overall goodness of fit for the final models were good (for boys: CFI = 0.960, TLI = 0.930, RMSEA = 0.0434, for girls: CFI = 0.968, TLI = 0.938, RMSEA = 0.044). Due to having a combination of categorical and discrete measures in the models, we used weighted least squares mean and variance estimators (WLSMV) to estimate the simultaneous equations. The method handles missing data by using available data to estimate model parameters. Analyses were carried out in Mplus 8.6 software package (Muthen & Muthen, Los Angeles, CA 90066).

## Results

The total number of learners enrolled in grade 8 in the 24 schools in 2014 when recruitment was done was 6076. A total of 4095 learners had parental consent to participate in the study, and 3811 gave learner assent, agreeing to participate. Of the 3811, 3743 completed the baseline survey: 2118 girls (56.6%) and 1625 boys (43.4%). Fourteen percent of boys and six percent of girls were 15 years and older. Most of the learners (69%) lived in a brick house and almost half

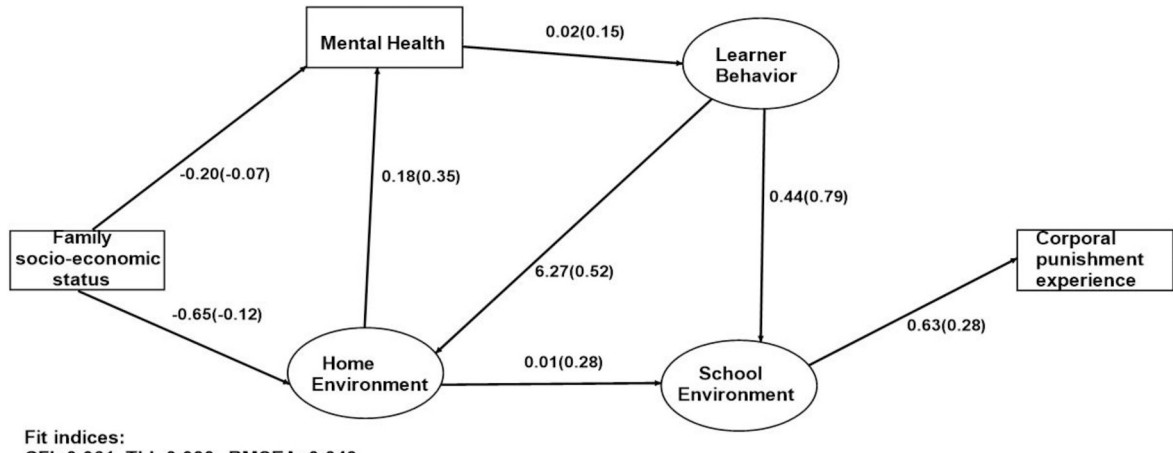

**Fig 2. Pathways to experiencing corporal punishment at school among boys.**

(46%) lived with both biological parents. One in seven learners (14.5%) did not live with their biological parent. Forty-seven percent of the learners were living with an unemployed care-giver. More boys (47%) than girls (39%) belonged to a club or society. Most girls (90%) were actively involved in church compared to 85% of boys. Sixty percent of learners had ever dated, and six percent had used alcohol and drugs. Boys were more likely to have dated, to use alcohol or other substances (**Table 2**).

One in five learners (20.6%) had repeated a grade, and boys (28.1%) were more likely to have done so compared to girls (14.9%). Fifty-two percent of learners had experienced CP from teachers at school, and 44.8% from parents or caregivers at home in the last 6 months with levels of experience higher among boys compared to girls. More than half (60.5%) of boys had experienced CP at school and 48.3% at home, while 46.3% of girls experienced CP at school and 42.3% at home in the 6 months preceding the interview.

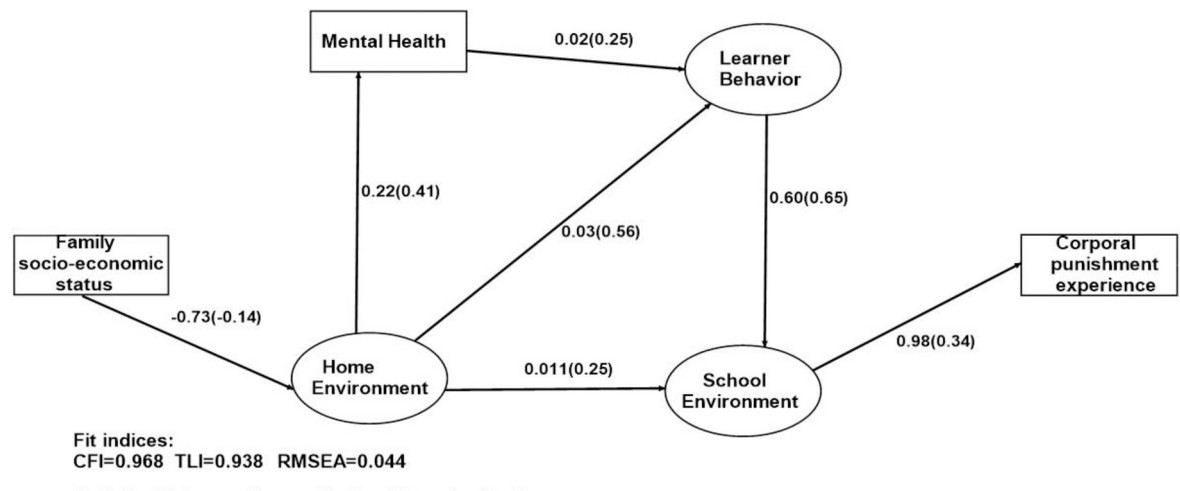

**Fig 3. Pathways to experiencing corporal punishment at school among girls.**

**Table 2. Sample characteristics and school corporal punishment experience by sex of learners.**

| | ALL | BOYS | | GIRLS | |
|---|---|---|---|---|---|
| | N | n | % | n | % |
| **Age of learner** | | | | | |
| < = 12yrs | 790 | 248 | 15.26 | 542 | 25.59 |
| 13yrs | 1871 | 739 | 45.48 | 1132 | 53.45 |
| 14yrs | 728 | 408 | 25.11 | 320 | 15.11 |
| > = 15yrs | 354 | 230 | 14.15 | 124 | 5.85 |
| **Type of housing** | | | | | |
| Brick house/flat | 2587 | 1128 | 69.5 | 1459 | 68.92 |
| Wendy/cottage | 601 | 263 | 16.2 | 338 | 15.97 |
| Informal settlement | 552 | 232 | 14.29 | 320 | 15.12 |
| **Race** | | | | | |
| Black | 3437 | 1480 | 91.4 | 1957 | 92.5 |
| Other | 297 | 139 | 5.6 | 158 | 7.5 |
| **Club/society membership** | 1585 | 758 | 46.76 | 827 | 38.99 |
| **Active church member** | 3283 | 1371 | 84.53 | 1912 | 90.23 |
| **Biological mother main woman in house** | 2869 | 1228 | 75.66 | 1641 | 77.41 |
| **Biological father main man in house** | 2035 | 929 | 57.27 | 1106 | 52.24 |
| **Biological parents at home:** | | | | | |
| None | 541 | 227 | 14.02 | 314 | 14.83 |
| Mother only | 1339 | 547 | 33.79 | 792 | 37.39 |
| Father only | 142 | 69 | 4.26 | 73 | 3.45 |
| Both father & mother | 1715 | 776 | 47.93 | 939 | 44.33 |
| **Caregiver employment** | | | | | |
| None | 1745 | 800 | 49.41 | 945 | 44.62 |
| Female caregiver only | 856 | 372 | 22.98 | 484 | 22.85 |
| Male caregiver only | 546 | 223 | 13.77 | 323 | 15.25 |
| Both male and female caregiver | 590 | 224 | 13.84 | 366 | 17.28 |
| **Ever-dated** | 2249 | 1140 | 70.72 | 1109 | 52.46 |
| **Repeated Grade** | 770 | 455 | 28.12 | 315 | 14.89 |
| **Substance or alcohol use** | 212 | 137 | 8.5 | 75 | 3.55 |
| **Experienced corporal punishment at school in past 6 months** | 1953 | 974 | 60.5 | 979 | 46.33 |
| **Experienced physical punishment at home in past 6 months** | 1675 | 781 | 48.27 | 894 | 42.25 |

Table 3 shows the relationship between experiencing CP at school and individual measures of the different latent constructs. Experiencing CP at school was associated with all individual measures of learner behavior (substance use, sexual behavior, and misbehaviour), and with all individual measures of home environment (caregiver communication score, caregiver kindness score, corporal punishment score and learner's neglect score), for both boys and girls. Less communication between caregiver and learner was associated with increased risk of experiencing corporal punishment. Caregiver unkindness was associated with increased risk of experiencing corporal punishment. Similarly, learners who experienced physical punishment at home were more likely to experience corporal punishment at school. Being neglected at home was associated with increased risk of experiencing corporal punishment at school. There were significant relationships between experiencing CP at school and individual measures of caregiver attitude to school work amongst girls. Caregiver's negative attitude towards learner's school work was associated with learner's experience of corporal punishment at school. Caregiver's attitude to school work was not associated with experiencing CP amongst

**Table 3. Bivariate logistic regression of learners' experience of corporal punishment at school and their behavior, home and school environment.**

| | BOYS | | | | GIRLS | | | |
|---|---|---|---|---|---|---|---|---|
| | No corporal punishment | Corporal punishment | | | No corporal punishment | Corporal punishment | | |
| **Learner's behavior** | **Mean score** | **Mean score** | **OR(95%CI)** | **p-value** | **Mean score** | **Mean score** | **OR(95%CI)** | **p-value** |
| Substance use score (high = more use) | 0.57 | 0.90 | 1.15(1.04–1.28) | 0.009 | 0.31 | 0.48 | 1.17(1.09–1.26) | <0.001 |
| Misbehaviour score(high = more misbehaviour) | 6.08 | 6.89 | 1.15(1.10–1.21) | <0.001 | 5.67 | 6.06 | 1.15(1.09–1.20) | <0.001 |
| Sexual behavior score (high = more involved) | 2.45 | 3.13 | 1.11(1.07–1.15) | <0.001 | 1.53 | 1.8 | 1.37(1.18–1.60) | <0.001 |
| **School environment** | | | | | | | | |
| Caregiver attitude to school (high = negative attitude) | 1.59 | 1.59 | 1.00(0.87–1.14) | 0.962 | 1.41 | 1.56 | 1.23(1.11–1.36) | <0.001 |
| Learner's perception of teacher behavior score (high = negative perception) | 19.28 | 21.42 | 1.06(1.04–1.08) | <0.001 | 18.3 | 20.4 | 1.07(1.05–1.09) | <0.001 |
| Learner's bullying score (high = more bullying) | 5.51 | 5.86 | 1.06(1.02–1.10) | 0.005 | 4.82 | 5.23 | 1.09(1.04–1.13) | <0.001 |
| **Home environment** | | | | | | | | |
| Caregiver kindness score (high = unkind) | 11.64 | 12.24 | 1.03(1.01–1.05) | 0.016 | 11.66 | 12.35 | 1.03(1.01–1.05) | 0.001 |
| Caregiver communication score (high = poor communication) | 7.57 | 7.96 | 1.03(1.01–1.06) | 0.024 | 7.09 | 7.88 | 1.06(1.04–1.09) | <0.001 |
| Physical punishment at home score (high = more punishment) | 1.27 | 2.07 | 1.15(1.10–1.21) | <0.001 | 1.21 | 1.98 | 1.15(1.11–1.19) | <0.001 |
| Neglect (high = more neglect) | 2.86 | 3.14 | 1.11(1.04–1.20) | 0.004 | 2.48 | 2.69 | 1.15(1.07–1.23) | <0.001 |
| **Other factors** | | | | | | | | |
| Socio-economic status score (high = better SES) | 6.26 | 6.05 | 0.93(0.86–1.01) | 0.092 | 6.14 | 5.95 | 0.94(0.88–0.99) | 0.038 |
| Depression score (high = more depression) | 3.8 | 4.66 | 1.04(1.02–1.06) | <0.001 | 3.71 | 4.31 | 1.03(1.01–1.05) | 0.004 |

boys. High household SES was associated with decreased risk of CP experience while depression was associated with increased risk of CP experience.

**Fig 2** and **Table 4** shows the pathways leading to boys' experience of CP at school. The structural equation model shows a direct effect of family SES on the learner's home environment, with low family SES status associated with negative home environment ($\beta$ = -0.12, p-value<0.001). Low family SES status also has direct negative impact on a learner's mental health ($\beta$ = -0.07, p-value = 0.012). The home environment has direct impact on learner's mental health ($\beta$ = 0.35, p-value<0.001) and their behavior at school (to school work or towards educators or peers) ($\beta$ = 0.28, p-value<0.001). The model also shows a significant impact of learner's mental health on their general risky behavior (substance use, misbehaviour) ($\beta$ = 0.15). For the boys, their misbehavior affects their relationship with their parents or caregivers at home ($\beta$ = 0.52, p-value<0.001). As hypothesized, learner's experience of CP at school is influenced mainly by their school environment (teacher behavior, learner's behavior towards teacher's and other learners, and parents or caregiver's attitude towards schoolwork) ($\beta$ = 0.28, p-value<0.001).

**Fig 3** and **Table 4** show the pathways leading to girls' experience of CP at school. The pathways are similar to boys as was hypothesized but with some minor differences. For girls, despite showing negative impact of low family SES on mental health, this relationship was not

**Table 4. Pathways to experiencing corporal punishment at school for boys and girls.**

| Path | BOYS | | | GIRLS | | |
|---|---|---|---|---|---|---|
| | Unstd.Coef(95%CI) | Std.Coef. | p-value | Unstd.Coef(95%CI) | Std.Coef. | p-value |
| Family socio-economic status→ Home environment | -0.65(-0.78 - -0.53) | -0.12 | <0.001 | -0.73(-0.85 - -0.61) | -0.14 | <0.001 |
| Family socio-economic status→ Mental Health | -0.20(-0.36 - -0.05) | -0.07 | 0.012 | | | |
| Home environment→ Mental Health | 0.18(0.11–0.25) | 0.35 | <0.001 | 0.22(0.14–0.31) | 0.41 | <0.001 |
| Mental Health → Learner behavior | 0.02(0.01–0.04) | 0.15 | 0.001 | 0.02(0.016–0.026) | 0.25 | <0.001 |
| Learner behavior → Home environment | 6.27(3.20–9.34) | 0.52 | <0.001 | | | |
| Home environment → Learner behavior | | | | 0.03(0.02–0.04) | 0.56 | <0.001 |
| Home environment → School environment | 0.01(0.01–0.02) | 0.28 | 0.001 | 0.011(0.003–0.018) | 0.25 | 0.005 |
| Learner behavior → School environment | 0.44(0.31–0.56) | 0.79 | <0.001 | 0.60(0.45–0.75) | 0.65 | <0.001 |
| School environment → Experiencing corporal punishment | 0.63(0.45–0.81) | 0.28 | <0.001 | 0.98(0.77–1.18) | 0.34 | <0.001 |
| **ii) Variances** | | | | | | |
| Mental Health | 20.9(19.3–22.4) | 0.83 | | 16.8(15.8–17.7) | 0.84 | |
| Home environment | 62.9(7.5–118.4) | 0.68 | | 64.5(18.2–110.8) | 0.98 | |
| School environment | 0.01(-0.03–0.05) | 0.05 | | 0.04(0.02–0.06) | 0.30 | |
| Learner behavior | 0.59(0.46–0.72) | 0.93 | | 0.07(0.06–0.09) | 0.51 | |
| **iii) R-squared for Latent factors** | | | | | | |
| Home environment | 0.322 | | | 0.022 | | |
| School environment | 0.951 | | | 0.697 | | |
| Learner behavior | 0.072 | | | 0.488 | | |

Unstd.Coef = Unstandardized coefficients. Std.Coef = Standardized coefficients.

significant. But consistent with boys, there was a strong relationship between home environment and learner's mental health (β = 0.41, p-value<0.001), learner's school climate (teacher behavior, learner's behavior towards teachers and other learners, and parents or caregiver's attitude towards schoolwork) (β = 0.25, p-value = 0.005), and their misbehavior (β = 0.56, p-value<0.001). Boys mental health has impact on their general misbehavior (substance use and general misbehaviour) (β = 0.15, p-value<0.001). As hypothesized and consistent with boys, girls' experience of CP at school is influenced mainly by their school environment (teacher behavior, learner's behavior towards teachers and other learners, and caregiver's attitude towards schoolwork) (β = 0.34, p-value<0.001).

## Discussion

This study aimed to describe the prevalence, factors associated with, and pathways to experiencing CP at school in the past six months. We found a high prevalence of CP experienced by learners at school, with more than half (52%) of the learners reporting CP experience in the past six months. The findings of this study highlight a slightly higher prevalence compared to that found in a national study (49.8%) conducted in 2012 [10], and confirm that CP of learners in public schools persists, despite it being outlawed over 20 years ago [13, 25]. Boys were more likely to experience CP at school compared to girls. This finding is consistent with findings of research from other countries which have shown gender differences in the experience of CP at school [16, 33]. Boys are perceived as naughty and mischievous compared to girls, which explains why they are more likely to experience CP [34, 35].

Our data has shown that the risk factors associated with learner experience of CP at school includes learner's behaviour, home environment, school climate, and other factors including family's SES and learner's mental health. For learners, irrespective of gender, school

environment (i.e. caregiver's attitude towards school work, learner's perception of teacher behaviour and learner bullying score) has a direct association with learner experience of CP. Learners' negative perceptions towards teacher behaviour were associated with experiencing CP at school. Current evidence suggests that this relationship could go either direction. For example, some studies have shown that, for learners, having negative perceptions towards teacher behaviour increases the risk of experiencing CP at school, whereas others have shown that CP at school leads to learners' negative perception towards teacher behaviour [36, 37]. The cross-sectional nature of the data analysed was a limitation to establishing temporality. However, our use of structural equation modelling enabled us to establish the pathways and inter-relationships of variables.

Even though in the bivariate analysis we found an association between sexual behaviour, substance use, bullying and CP at school; overall, no direct association was found between learner behaviour and CP at school in the multivariate analysis. For boys, learner behaviour had direct effects on home and school environment while among the girls, learner behaviour had direct effects on school environment only. School environment was the key factor associated with experience of CP for boys and girls, and this relationship was mediated by other factors including learner behavior and home environment. Addressing learner experience of CP at school requires that we not only focus on learner behaviour, but also on the home and school environments which have a direct association. As hypothesized, our data have shown that the home environment (exposure to physical violence, neglect and lack of caregiver kindness and support) has an indirect relationship with experience of CP at school, mediated by the school environment. Amongst girls, the home environment has a direct association with learner behaviour, which has an influence on the school environment, and experience of CP at school. This finding can be explained by the gender role expectations amongst girls compared to boys in some South African homes [38]. Some caregivers do not prioritize nor support education of girls who are mainly occupied by household chores in the home [39]. Young girls are socialised to clean, cook, take care of the children and to manage other household responsibilities in the home compared to boys [38, 40]. As such, most girls spend more time doing household chores than school-work when they are at home [39, 41]. Furthermore, the lack of caregiver support, and experience of neglect can contribute to misbehaviour which attracts CP at home and school. For many children, home life that is characterized by a parenting style which supports the use of CP as an effective method of managing behaviour is more likely to spillover to the school, with some parents or caregivers expecting teachers to use CP at school [4, 11, 13].

Studies have shown that boys are more likely to misbehave which results in intolerance and negative caregiver and teacher attitude towards them [42, 43]. With lack of support from both the parent or caregiver and the teacher, boys are more likely to perform poorly at school, with the poor performance attracting experience of CP [17, 44]. The association between lack of parental or caregiver involvement and poor academic performance in learners has been demonstrated in the literature [45–47]. Experience of school CP amongst boys could be aimed at correcting their behaviour, based on the belief that physical punishment is effective in correcting deviant behaviour of children [4]. However, there is no consensus in the literature about the effectiveness of CP as a deterrent of misbehavior amongst learners. A study conducted in South African high schools in Soweto found that learners had become so insensitive to the physical pain inflicted through CP that their misbehavior was exacerbated rather than being curbed by it (Ngubane et al., 2019). The intolerance of boys' misbehaviours could be emanating from caregivers' and teachers' limited capacity to manage misbehavior in children [48]. Corporal punishment could be the only corrective strategy familiar to caregivers at home, and teachers at school. As CP is illegal both in the home and at school in South Africa, programs

for caregivers and teachers aimed to build capacity on using positive disciplining strategies are needed to end the use of CP [3, 49].

Furthermore, our findings have shown that the home environment was directly associated with learner's mental health and learner behavior amongst girls. Learners who experience lack of care and support, and neglect at home are likely to experience poor mental health, which contributes to learner misbehaviour. Our study has shown that the influence of home environment and mental health also connects through family SES amongst boys. A gender effect of socio-economic status was also observed in other studies [50]. Interventions that address poor mental health are important for developmental wellbeing and academic performance of learners. A number of studies have highlighted the importance of caregiver involvement in improving both academic and mental health outcomes amongst adolescents [51–53]. A positive family environment, and caregiver emotional support is associated with improved educational outcomes, social functioning and coping amongst adolescents [53, 54]. Adolescents with mental well-being possess problem-solving skills, social competence and a sense of purpose, which makes them resilient and more able to thrive in adverse circumstances [52].

The study had limitations in that the findings only represents the 24 schools included in the research, and cannot be generalized to other schools in Gauteng. Data used in this analysis formed part of the baseline assessment of the Skhokho RCT, thus other variables that are documented in the literature, and found to be associated with experience of CP such as neighbourhood and community level factors were not measured. Furthermore, the analysis was conducted from quantitative baseline data. A qualitative exploration of the topic might have provided further details about learner experiences of corporal punishment in schools. Given that use of CP is a behaviour of teachers, it is important to explore factors associated with teachers use of CP. Future qualitative research with learners and teachers is warranted to strengthen our understanding of not only the learners' experience of CP but also teachers' insights on why CP continues to be used at school, despite the legislation prohibiting its use.

## Conclusion

Our findings have shown that learners still experience CP in public schools in South Africa. While there is a need to also focus on learner behaviour, our findings have highlighted the need to intervene in both the home and school environments. There is an urgent need to break this cycle of violence, through use of CP, by enforcing the law, and holding accountable those who continue to use CP despite legislation prohibiting its use, but also supporting the children in their home environment, and their parents to positively partner with their children. Furthermore, there is a need to provide services to meet leaner's mental health needs. Evidence-based interventions are needed to support both parents and teachers in managing learner behaviour. Use of positive disciplining strategies, developing democratic relationship and consciousness about image of a child can positively informt how parents and teachers relate with children, critical for raising responsible children, and to curb future perpetration of violence in society [55, 56]. Effecting engrained beliefs, attitudes and behaviours supportive of corporal punishment amongst parents and teachers will require consistent education about the harms of corporal punishment and support for why it needs to be ended. Existing life skills curricula in schools need to be enhanced to assist learners with skills to deal with mental health issues and to build their resilience to withstand socio-economic hardships.

## Acknowledgments

We thank the Gauteng Department of Basic Education for giving us permission to conduct the study, the National Department of Basic Education for support with the research; the

Principals who gave us access to schools, the caregivers, teachers and children who participated; members of our stakeholder advisory committee; our first project manager Jacqueline Mangoma-Chaurura and all the project staff.

## Author Contributions

**Conceptualization:** Pinky Mahlangu, Esnat Chirwa, Nwabisa Shai, Rachel Jewkes.

**Data curation:** Esnat Chirwa.

**Formal analysis:** Esnat Chirwa, Rachel Jewkes.

**Funding acquisition:** Rachel Jewkes.

**Investigation:** Pinky Mahlangu.

**Methodology:** Nwabisa Shai, Rachel Jewkes.

**Project administration:** Pinky Mahlangu.

**Writing – original draft:** Pinky Mahlangu, Esnat Chirwa.

**Writing – review & editing:** Pinky Mahlangu, Esnat Chirwa, Mercilene Machisa, Yandisa Sikweyiya, Nwabisa Shai, Rachel Jewkes.

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
