## [Decision Letter · Decision Letter 0]

1 Mar 2021

PONE-D-20-20199

Prevalence and factors associated with experience of corporal punishment in public schools in South Africa

PLOS ONE

Dear Dr. Mahlangu,

Thank you for submitting your manuscript to PLOS ONE. After careful consideration, we feel that it has merit but does not fully meet PLOS ONE’s publication criteria as it currently stands. Therefore, we invite you to submit a revised version of the manuscript that addresses the points raised during the review process.

We look forward to receiving your revised manuscript.

Kind regards,

Thach Duc Tran, M.Sc., Ph.D.

Academic Editor

PLOS ONE

Journal Requirements:

2. Please address the following:

- Please include additional information regarding the survey or questionnaire used in the study and ensure that you have provided sufficient details that others could replicate the analyses. For instance, if you developed a questionnaire as part of this study and it is not under a copyright more restrictive than CC-BY, please include a copy, in both the original language and English, as Supporting Information.

- In the ethics statement in the Methods and online submission information, please ensure that you have specified whether parental consent was informed.

3.In your Data Availability statement, you have not specified where the minimal data set underlying the results described in your manuscript can be found. PLOS defines a study's minimal data set as the underlying data used to reach the conclusions drawn in the manuscript and any additional data required to replicate the reported study findings in their entirety. All PLOS journals require that the minimal data set be made fully available. For more information about our data policy, please see http://journals.plos.org/plosone/s/data-availability.

5. Thank you for stating in your financial disclosure: 

"RJ received funding from an anonymous

donor who had no role in the study; RJ received funding from the South African Medical Research Council. These sponsors played no role in the research."

PLOS ONE requires you to include in your manuscript further information about the funder so that any relevant competing interests can be assessed. Please respond to the following questions:

a) Please state whether any of the research costs or authors' salaries were funded, in whole or in part, by a tobacco company (our policy on tobacco funding is at http://journals.plos.org/plosone/s/disclosure-of-funding-sources) 

b) Please state whether the donor has any competing interests in relation to this work (see http://journals.plos.org/plosone/s/competing-interests) .

c) Please state whether the identity of the donor might be considered relevant to editors or reviewers’ assessment of the validity of the work.

d) If the donors have no perceived or actual competing interests, please state: “The authors are not aware of any competing interests”.

This information should be included in your cover letter. We will amend your financial disclosure and competing interests on your behalf.

Reviewers' comments:

Reviewer's Responses to Questions

**Comments to the Author**

1. Is the manuscript technically sound, and do the data support the conclusions?

Reviewer #1: Yes

Reviewer #2: Yes

Reviewer #3: Yes

2. Has the statistical analysis been performed appropriately and rigorously? 

Reviewer #1: I Don't Know

Reviewer #2: I Don't Know

Reviewer #3: Yes

3. Have the authors made all data underlying the findings in their manuscript fully available?

Reviewer #1: Yes

Reviewer #2: Yes

Reviewer #3: Yes

4. Is the manuscript presented in an intelligible fashion and written in standard English?

Reviewer #1: Yes

Reviewer #2: Yes

Reviewer #3: Yes

5. Review Comments to the Author

Reviewer #1: Important note: This review pertains only to ‘statistical aspects’ of the study and so ‘clinical aspects’ [like medical importance, relevance of the study, ‘clinical significance and implication(s)’ of the whole study, etc.] are to be evaluated [should be assessed] separately/independently. Further please note that any ‘statistical review’ is generally done under the assumption that (such) study specific methodological [as well as execution] issues are perfectly taken care of by the investigator(s). This review is not an exception to that and so does not cover clinical aspects {however, seldom comments are made only if those issues are intimately / scientifically related & intermingle with ‘statistical aspects’ of the study}. Agreed that ‘statistical methods’ are used as just tools here, however, they are vital part of methodology [and so should be given due importance].

COMMENTS: Abstract-Methods section says “3743 grade 8 learners (2118 girls and 1625 boys) from 24 selected public schools in Tshwane, South Africa, enrolled in a cluster randomised controlled trial” my question is if the schools (clusters) are purposively selected, where is randomization? [according to lines 109-111, “All Grade 8 learners in 24 purposively selected English medium State secondary schools within a 50km radius from Pretoria City were eligible to participate in the study”].

Although, it is said in line 178 that [‘All data analysis procedures used took into account the study design’], it is not observed. In this context [authors may already know them, however, not found in the references of this manuscript], here are three good references.

1. Donner Allen and Klar Neil. `Design and Analysis of Cluster Randomization Trial in Health Research’, Oxford University Press Inc., New York, 2000.

2. Bland JM and Kerry SM. ‘Statistics notes: trials randomized in clusters’, Br. Med. Jr., 1997, Volume 315, p600.

3. Kerry SM & Bland JM. ‘Analysis of a trial randomized in clusters’ by in British Medical Journal, Volume 316, 1998, p54.

Also note that

Though the measures/tools used are appropriate [Example - Line 112-3: A survey was conducted using a self-administered questionnaire loaded in personal digital assistants (PDAs)], most of them yield data that are in [at the most] ‘ordinal’ level of measurement [and not in ratio level of measurement for sure {as the score two times higher does not indicate presence of that parameter/phenomenon as double (for example, a Visual Analogue Scales VAS score or say ‘depression’ score)}]. Then application of suitable non-parametric test(s) is/are indicated/advisable [even if distribution may be ‘Gaussian’ (i.e. normal)]. Agreed that there is/are no non-parametric test(s)/technique(s) available to be used as alternative in all situation(s) [suitable / most desired/applicable], but should be used wherever/wherever they are available.

From Table 1 [Description of the factors that influence experience of corporal punishment at school] it appears that many measures used are of this type. I request the authors to also note that (this note is copied and pasted from a standard text book) : Whenever response options ranged from 1=strongly disagree to 4=strongly agree (or ranging from 1 (strongly disagree) to 6 (strongly agree) or from 1=very bad to 3=neither good nor bad to 5=very good), while using a ‘Likert’ scale responses, recoding [like strongly disagree=-2, disagree=-1, neutral=0, agree=1, strongly agree=2] may yield correct and meaningful ‘arithmetic mean’ which is useful not only for comparison but has absolute meaning, in my opinion. Application of any statistical test(s) assume that meaning of entity used (mean, SD, etc) has a particular meaning. Though ‘α’ [alpha] or most other measures of reliability/correlation will remain same, however. Use of non-parametric methods should/may be preferred while dealing with data yielded by any questionnaire/score.

From account given in lines 228-9 [Table 3 shows the relationship between experiencing CP at school and individual measures of the different latent constructs] it is clear that in table-3 {Bivariate relationship between corporal punishment experience score and a learner’s home environment, attitude and behavior measures.} first column Coef is ‘Correlation Coefficient’ (mostly Pearson’s) {LCL is lower & UCL is upper limit of CI I guess, why not given in footnote). Coefficient values are very small, yet most p-values show highly significant results. Is not that contradictory? In this context again it may please be noted that (following note is from a standard text book):

Statistical test usually used to assess significance of Pearson’s ‘Correlation coefficient (r)’ is ‘t’ [where t = { r � [(n-2) / (1-r2)] }for df=n-2, n is sample size] and here Ho is that the population/standard value of ‘r’ is zero. You need r=0.878 to be significant at 5% when n=5 but you need r=0.273 if n=50 & you need only r=0.088 if n=500. ‘P-value’ heavily depends on sample size. Therefore, it is customary to use the (available in most text books on ‘Biostatistics’ or on ‘www/net’) guidelines [very strongly suggesting to consider an absolute value of ‘Correlation coefficient’] for interpreting positive or negative correlations (and do not rely only on corresponding ‘P’-value but also consider an absolute value of ‘Correlation coefficient’). [This argument is equally applicable to non-parametric Spearman’s ‘Correlation coefficient (ρ)’ as well.]

Please look at an absolute value of ‘Correlation coefficient’ in table-3. It is suggested that ‘If r = +.70 or higher Very strong positive relationship’ or ‘-.70 or higher Very strong negative relationship’. In table-3, all ‘Coef’ are below 0.18 [smallest as small as 0.02]. It is not mentioned however that which type of correlation coefficient is used. But in case in the light of the above, interpretation given in lines 228 onwards is questionable.

Moreover, as said in lines 140-145 CP at school is measured by score [It is derived from 2 items: (i) ‘In the past 6 months were you ever beaten by a teacher?’, with a ‘yes or no’ response and (ii) “A teacher might beat you or physically punish you at our school” with responses as strongly disagree, disagree, agree and strongly agree. Our exploratory analysis showed that learners who agreed to statement (ii) were also likely to respond “yes’ to the first statement. We thus created a score from the 2 responses where high score represents higher likelihood of experiencing CP.] and both ‘simple linear regression’ as well as ‘correlation’ {lines 181-3: We used simple linear regression to assess the relationship between experience of CP at school and all individual measures described in Table 1, and also assessed the correlation amongst the measures} assume that (both ‘independent’ & ‘dependent’) variables are continuous.

Surprisingly use of ‘structural equation modelling’ seems to be correct. Account given in lines 184-198 seems perfectly alright. Figures 1,2,3 are alright. Look at the magnate of coefficients in contrast to coefficients reported/displayed in table-3, though they may be different [in nature & purpose].

Reviewer #2: This is an interesting quantitative study on the prevalence of corporal punishment amongst 8 graders in 24 selected government schools in South Africa. The study is trying to uncover the underlying reasons for its prevalence and suggests a solution to it.

The article is well-written and makes a good contribution to the field in that it provides evidence for the fact that corporal punishment is happening. This is in itself important as research in CP is notoriously difficult, because CP is illegal since 1997 and could cause major problems for teachers if found out. It is interesting that this part of the study is not really discussed at all. Research ethics and what exactly was said to the school principals is missing. For a study of this kind it is important to also publish the letters of consent and assent and how some of that was negotiated. CP tends to go underground when there is a risk that teachers might get reported. But there is another issue worth mentioning and that is, if corporal punishment is defined as physical punishment, then why are the main questions of the survey phrased in terms of ‘Were you beaten…?’. From my experiences of researching CP, beatings are a small part of the abuse that is routinely going on in schools. I would really like to see the full set of questions used in the study. The evidence sought was to identify the risk factors. There is interesting info about the characteristics of children who receive more CP in the literature, which is again confirmed by this study. A qualitative component probably would have given more unexpected results, because that is often ‘hiding’ in the details. I would very much like to have learned about teachers’ perspectives, but how to conduct such a study (and get indeed genuine information) is another matter. However, I understand such a follow-up study is in the pipeline.

The research was carried out by the SA Medical Research Council and it is within that context that some of my comments might be helpful. I am not an expert in quantitative studies and this one doesn’t use a qualitative component (although it recommends it as a follow-up study), so I can’t comment on the use of the methods that were used, but the following might be helpful:

1. Line 26: To my knowledge CP is not prevalent in schools globally, maybe homes? It would have been helpful to be more specific as it is certainly not common in schools in Europe, Australia etc, although it might be normal in households as parents are often allowed to use CP but not teachers. The details are important here.

2. Page 4: Factors that influence the use of CP in US are of use, but it would have been helpful to hear more about the racialised discourses that inform the use of CP - e.g. children are seen as wild and need to be tamed, domesticated and controlled. Innocence tends not to be attributed to brown bodies – a discourse also internalised by black people themselves. It was striking to see a lack of reference to racial, ethnic, religious and other such factors that are critical in relationships in SA. I would really like to see all research instruments included as Appendices.

3. At times it isn't clear to which part of the world the info applies and claims are made. For example, first there is evidence from Pakistan, then later US, but on page 5 it is unclear whether the authors make claims about CP as a global phenomenon or not. The info needs to be tied more specifically to the geopolitical location. What is missing is background information about the racialised and historically complex two-tier education system. Classrooms are very large and on the whole teachers not as qualified as in many other places. Therefore, this article could unfairly maybe give a very negative impression of a terribly underpaid and under supported workforce. Teachers in SA work under very difficult conditions and this article should include more information about teachers’ conditions so it doesn’t run the risk of being accused of a deficit approach, especially as teachers’ voice in the study is absent. It is only learners’ perspectives.

4. Line 90: This is a generalisation not supported by evidence. Education departments in SA Universities seem to be doing their upmost to teach their students alternative discipline strategies, but the phenomenon is more complex than that. Also Provincial education departments all have initiatives to support teachers in using different discipline methods, but the particular image of child and childhood and child/adult relations tend to be hierarchical in SA and lack the equality needed for respectful non-violent relations. This is typical of patriarchal societies. The violence against children in SA is intricately connected to the violence against women.

5. What is completely missing is the relationship between the training of teachers, their ability to teach in ways that engages learners and motivates them. Teacher educators know that e.g. more active and embodied pedagogies can be used that prevent 'misbehaviour' from occurring. Large class sizes etc also really contribute to CP and not having classroom assistants.

6. The intervention focuses on positive discipline and psychological interventions, but not on the image of the child or the pedagogical dimensions. So in a sense, the hypothesis of the study makes sense in the light of the identity of the researchers. It would be helpful to include in future research someone from a very different educational background so base-line hypotheses can be enriched. We know from educational research that teachers tend to teach how they were taught and it is very difficult to break through that despite the many interventions in teacher education and by provincial departments. Therefore, the suggestion that the solution to the problem identified might be to offer training in positive discipline is nothing new and has been going on for some time now. It is a good idea to check out this literature. What would be worthwhile and unusual and new instead is to include an in-service course/intervention in childhood studies (to teach about the image of the child) and the building of democratic relationships through e.g. ‘shared authority’ models. These are about more sustainable and durable changes as they could affect deeply engrained beliefs about children as found wanting (by adults), rather than interventions that don’t change fundamental beliefs about people and their relations. However, this would require pedagogical skills and working with both groups at the same time: learners and their teachers.

In short, this is a very informative article, well written and researched and helpful in that it confirms much of what we already know about CP, and indeed helpful for funding purposes, but it misses the important geopolitical educational dimension (which still can be added) and references to pedagogies which hopefully can be taken into account in further studies. I hope the above comments are helpful, in order to add further reflections to the current paper, but especially I hope they are useful for future research purposes.

Reviewer #3: This study examines the prevalence and factors associated with corporal punishment in public schools in South Africa using baseline data from a larger RCT study.

This is a highly competent and well published team who have completed most of the seminal work on interpersonal violence in South Africa and in fact globally

The methodology of the larger randomized controlled trial is excellent and it appears to be a well conducted study

I have one analytic concern. The authors merge two questions ‘In the past 6 months were you ever beaten by a teacher?’ and “A teacher might beat you or physically punish you at our school”. These are two entirely different questions and simply because most people answer yes to the first have answered yes to the second (of course they have) is no reason to merge them. This must be corrected in the revised version.

The use of the term ‘delinquent’ is highly problematic, certainly not justified by the data collected in the paper and should be deleted. To describe a learner as delinquent based on 5 questions is highly problematic.

A more general concern has to do with what these findings add to the literature. Corporal punishment is associated with learner behaviour, home environment and school environment (i.e. everything). It does feel a little like squeezing another paper out of the dataset. While the paper adds little to what is already known, the method and execution of the study are of a high standard and the findings may be of some interest to education authorities in South Africa.

6. PLOS authors have the option to publish the peer review history of their article (what does this mean?). If published, this will include your full peer review and any attached files.

Reviewer #1: No

Reviewer #2: **Yes: **Karin Murris

Reviewer #3: No

---

## [Author Response · Author response to Decision Letter 0]

3 May 2021

Dear Editor

Thank you for the opportunity to revise our paper. Below we have described how we have responded to the comments from the reviewers

Best regards,

Reviewer 1 

Important note: This review pertains only to ‘statistical aspects’ of the study and so ‘clinical aspects’ [like medical importance, relevance of the study, ‘clinical significance and implication(s)’ of the whole study, etc.] are to be evaluated [should be assessed] separately/independently. Further please note that any ‘statistical review’ is generally done under the assumption that (such) study specific methodological [as well as execution] issues are perfectly taken care of by the investigator(s). This review is not an exception to that and so does not cover clinical aspects {however, seldom comments are made only if those issues are intimately / scientifically related & intermingle with ‘statistical aspects’ of the study}. Agreed that ‘statistical methods’ are used as just tools here, however, they are vital part of methodology [and so should be given due importance].

 Thank you very much for your time reviewing our manuscript and for the valuable feedback. We have revised and clarified the analytic methods applied to address issues raised, taking into account suggestions and recommendations from all the reviewers.

Abstract-Methods section says “3743 grade 8 learners (2118 girls and 1625 boys) from 24 selected public schools in Tshwane, South Africa, enrolled in a cluster randomised controlled trial” my question is if the schools (clusters) are purposively selected, where is randomization? [according to lines 109-111, “All Grade 8 learners in 24 purposively selected English medium State secondary schools within a 50km radius from Pretoria City were eligible to participate in the study”].

 While selection of the schools into the study was not random, the allocation of the selected schools to the interventions arms was random.

Although, it is said in line 178 that [‘All data analysis procedures used took into account the study design’], it is not observed. In this context [authors may already know them, however, not found in the references of this manuscript], here are three good references.

1. Donner Allen and Klar Neil. `Design and Analysis of Cluster Randomization Trial in Health Research’, Oxford University Press Inc., New York, 2000.

2. Bland JM and Kerry SM. ‘Statistics notes: trials randomized in clusters’, Br. Med. Jr., 1997, Volume 315, p600.

3. Kerry SM & Bland JM. ‘Analysis of a trial randomized in clusters’ by in British Medical Journal, Volume 316, 1998, p54.

 We acknowledge the source of information provided on analysis of cluster randomized. In our analysis, we performed individual level analysis considering:

i) We are analyzing baseline data for an outcome that is not primary trial outcome.

ii) We have a sizable number of clusters per arm and we accounted for the clustering 

in the estimation of standard errors in all the models.

Though the measures/tools used are appropriate [Example - Line 112-3: A survey was conducted using a self-administered questionnaire loaded in personal digital assistants (PDAs)], most of them yield data that are in [at the most] ‘ordinal’ level of measurement [and not in ratio level of measurement for sure {as the score two times higher does not indicate presence of that parameter/phenomenon as double (for example, a Visual Analogue Scales VAS score or say ‘depression’ score)]. Then application of suitable non-parametric test(s) is/are indicated/advisable [even if distribution may be ‘Gaussian’ (i.e. normal)]. Agreed that there is/are no non-parametric test(s)/technique(s) available to be used as alternative in all situation(s) [suitable / most desired/applicable], but should be used wherever/wherever they are available.

From Table 1 [Description of the factors that influence experience of corporal punishment at school] it appears that many measures used are of this type. I request the authors to also note that (this note is copied and pasted from a standard text book) : Whenever response options ranged from 1=strongly disagree to 4=strongly agree (or ranging from 1 (strongly disagree) to 6 (strongly agree) or from 1=very bad to 3=neither good nor bad to 5=very good), while using a ‘Likert’ scale responses, recoding [like strongly disagree=-2, disagree=-1, neutral=0, agree=1, strongly agree=2] may yield correct and meaningful ‘arithmetic mean’ which is useful not only for comparison but has absolute meaning, in my opinion. Application of any statistical test(s) assume that meaning of entity used (mean, SD, etc) has a particular meaning. Though ‘α’ [alpha] or most other measures of reliability/correlation will remain same, however. Use of non-parametric methods should/may be preferred while dealing with data yielded by any questionnaire/score.

From account given in lines 228-9 [Table 3 shows the relationship between experiencing CP at school and individual measures of the different latent constructs] it is clear that in table-3 {Bivariate relationship between corporal punishment experience score and a learner’s home environment, attitude and behavior measures. first column Coef is ‘Correlation Coefficient’ (mostly Pearson’s) {LCL is lower & UCL is upper limit of CI I guess, why not given in footnote). Coefficient values are very small, yet most p-values show highly significant results. Is not that contradictory? In this context again it may please be noted that (following note is from a standard text book):

Statistical test usually used to assess significance of Pearson’s ‘Correlation coefficient (r)’ is ‘t’ [where t = { r � [(n-2) / (1-r2)] }for df=n-2, n is sample size] and here Ho is that the population/standard value of ‘r’ is zero. You need r=0.878 to be significant at 5% when n=5 but you need r=0.273 if n=50 & you need only r=0.088 if n=500. ‘P-value’ heavily depends on sample size. Therefore, it is customary to use the (available in most text books on ‘Biostatistics’ or on ‘www/net’) guidelines [very strongly suggesting to consider an absolute value of ‘Correlation coefficient’] for interpreting positive or negative correlations (and do not rely only on corresponding ‘P’-value but also consider an absolute value of ‘Correlation coefficient’). [This argument is equally applicable to non-parametric Spearman’s ‘Correlation coefficient (ρ)’ as well.]

Please look at an absolute value of ‘Correlation coefficient’ in table-3. It is suggested that ‘If r = +.70 or higher Very strong positive relationship’ or ‘-.70 or higher Very strong negative relationship’. In table-3, all ‘Coef’ are below 0.18 [smallest as small as 0.02]. It is not mentioned however that which type of correlation coefficient is used. But in case in the light of the above, interpretation given in lines 228 onwards is questionable.

 We acknowledge the concern raised regards the distributional assumptions for the analytical methods used in the bivariate analysis. We do agree that non-parametric tests are more robust than parametric test for discrete measures.

Our aim for performing the correlations/linear regression analyses was mainly as an exploratory data analysis preceding the main analysis (structural equation model). This was done to show how the different measures were associated with the outcome (experience of corporal punishment), prior to the SEM analysis.

Based on recommendation from the review, we have defined our outcome measure (corporal punishment experience) as a binary outcome derived from responses from 1 question rather than from 2 questions as previously defined.

We have thus revised Table 3 to reflect this change and we now present relationships using Odds ratios from logistic regression models.

We have revised Tables

Moreover, as said in lines 140-145 CP at school is measured by score [It is derived from 2 items: (i) ‘In the past 6 months were you ever beaten by a teacher?’, with a ‘yes or no’ response and (ii) “A teacher might beat you or physically punish you at our school” with responses as strongly disagree, disagree, agree and strongly agree. Our exploratory analysis showed that learners who agreed to statement (ii) were also likely to respond “yes’ to the first statement. We thus created a score from the 2 responses where high score represents higher likelihood of experiencing CP.] and both ‘simple linear regression’ as well as ‘correlation’ {lines 181-3: We used simple linear regression to assess the relationship between experience of CP at school and all individual measures described in Table 1, and also assessed the correlation amongst the measures} assume that (both ‘independent’ & ‘dependent’) variables are continuous.

 As indicated above, we have redefined the outcome using only 1 item (In the past 6 months were you ever beaten by a teacher?)

Surprisingly use of ‘structural equation modelling’ seems to be correct. Account given in lines 184-198 seems perfectly alright. Figures 1,2,3 are alright. Look at the magnate of coefficients in contrast to coefficients reported/displayed in table-3, though they may be different [in nature & purpose]. Thank you. We have revised the SEM analysis and fitted a Generalised SEM with the experience of corporal punishment defined as a binary outcome.

Reviewer #2 

This is an interesting quantitative study on the prevalence of corporal punishment amongst 8 graders in 24 selected government schools in South Africa. The study is trying to uncover the underlying reasons for its prevalence and suggests a solution to it.

 Thank you very much for your time reviewing our manuscript and for the valuable feedback.

The article is well-written and makes a good contribution to the field in that it provides evidence for the fact that corporal punishment is happening. This is in itself important as research in CP is notoriously difficult, because CP is illegal since 1997 and could cause major problems for teachers if found out. It is interesting that this part of the study is not really discussed at all. Research ethics and what exactly was said to the school principals is missing. For a study of this kind it is important to also publish the letters of consent and assent and how some of that was negotiated. CP tends to go underground when there is a risk that teachers might get reported. But there is another issue worth mentioning and that is, if corporal punishment is defined as physical punishment, then why are the main questions of the survey phrased in terms of ‘Were you beaten…?’. From my experiences of researching CP, beatings are a small part of the abuse that is routinely going on in schools. I would really like to see the full set of questions used in the study. The evidence sought was to identify the risk factors. There is interesting info about the characteristics of children who receive more CP in the literature, which is again confirmed by this study. A qualitative component probably would have given more unexpected results, because that is often ‘hiding’ in the details. I would very much like to have learned about teachers’ perspectives, but how to conduct such a study (and get indeed genuine information) is another matter. However, I understand such a follow-up study is in the pipeline. a. The use of corporal punishment in schools is prohibited by law in South Africa. Any person who administers corporal punishment at school is guilty of an offence and liable on conviction to a sentence which could be imposed for assault. This has been added in the introduction section on page:

The South African Schools Act No 84 of 1996 states that no person may administer CP to a learner at school (1). Any person who administers corporal punishment at school is guilty of an offence and liable on conviction to a sentence which could be imposed for assault (1). 

b. We have added on page 9 line 138 a sub-heading ‘Ethical considerations. We have also added a paragraph in line 141 – 148 explaining what was said to principals when requesting permission to work in schools:

Permission was also granted by the school principals in the 24 schools. The information letter detailing the nature and purpose of the study, the interventions, participants, what the project would offer to schools, risks and benefits of participating in the study, project timelines, and on rights of participants was provided to principals. School principals were also informed that all data generated from the study will be kept confidential and that research reports and articles which will be submitted in scientific journals will not include any information that may identify the school or any of the educators, learners, school governing body officials and parents. 

c. We agree with the reviewer that CP tends to go underground, underreported when there is a risk that teachers might get reported. This is also shown in studies where both learners and teachers have been involved and the findings showed varied reports of use of corporal punishment in schools, learners reporting higher prevalence that teachers. In our study, use of corporal punishment in the 24 schools was reported by learners

d. Table 1 describes the main aspects of the questionnaire 

e. We used the Childhood Trauma Questionnaire and cognitively tested the questionnaire amongst grade 8 learners in schools which were not part of the research and found that ‘beaten’ was understood and captured corporal punishment as was intended

f. We have expanded the limitations section on page 27, line 439 - 445 to acknowledge the limitation of not including a qualitative component and teacher’s perspectives:

Furthermore, the analysis was conducted from quantitative baseline data. A qualitative exploration of the topic might have provided further details about learner experiences of corporal punishment in schools. Given that use of CP is a behaviour of teachers, it is important to explore factors associated with teachers use of CP. Future qualitative research with learners and teachers is warranted to strengthen our understanding of not only the learners’ experience of CP but also teachers’ insights on why CP continues to be used at school, despite the legislation prohibiting its use. 

The research was carried out by the SA Medical Research Council and it is within that context that some of my comments might be helpful. I am not an expert in quantitative studies and this one doesn’t use a qualitative component (although it recommends it as a follow-up study), so I can’t comment on the use of the methods that were used, but the following might be helpful:

 Thank you very much for your time reviewing our manuscript and for the valuable feedback

1. Line 26: To my knowledge CP is not prevalent in schools globally, maybe homes? It would have been helpful to be more specific as it is certainly not common in schools in Europe, Australia etc, although it might be normal in households as parents are often allowed to use CP but not teachers. The details are important here.

 While use of corporal punishment in school is banned and no longer an issue in European countries, it is a problem that requires attention in many other parts of the world. Corporal punishment is still prevalent in a third of the world’s countries where it continues to be used as a method of disciplining children, both in countries where it is legal and countries where it is banned, leading to estimates that millions of children are subject to this harmful practice (2, 3). 

We have revised the introduction section to become more specific about prevalence of CP globally, line 25 – 31:

While the use of a physical methods of discipline, also referred to as corporal punishment (CP) at school is banned and no longer an issue in European countries; it continues to be prevalent in a third of the world’s countries, despite evidence regarding its harmful physical, mental and behavioural effects on the child (1, 2). The global prevalence of CP in schools ranges between 13% – 97% of learners who reported experience of CP at school (1). Learners continue to experience CP in most countries in Sub-Saharan Africa despite legislation prohibiting its use (1, 2).

2. Page 4: Factors that influence the use of CP in US are of use, but it would have been helpful to hear more about the racialised discourses that inform the use of CP - e.g. children are seen as wild and need to be tamed, domesticated and controlled. Innocence tends not to be attributed to brown bodies – a discourse also internalised by black people themselves. It was striking to see a lack of reference to racial, ethnic, religious and other such factors that are critical in relationships in SA. I would really like to see all research instruments included as Appendices.

 We agree with the reviewer that race, ethnicity and religion are critical in relationships in South Africa. However, unlike in American schools, most public schools in South Africa are still very much racially segregated by communities served in that area. While the system of schools divided by race was eliminated in 1996s, public schools in Black townships or communities are racially homogenous, while the historically White and Indian schools serve a more racially diverse constituency (5, 6). 

About 21 of the schools where this research was conducted were black only schools: Only three of the 24 schools were mixed race schools. Even in the three mixed race schools, more than 90% of the students were black and this is shown in Table 2 on page 16

The reviewer is correct about the diversity in religion and ethnicity in schools in South Africa, recognized in the National Policy on Religion and Education of 2003. It is possible that religion and ethnicity could have been amongst the factors associated with learner experience of corporal punishment in schools, but we did not include in our analysis. 

3. At times it isn't clear to which part of the world the info applies and claims are made. For example, first there is evidence from Pakistan, then later US, but on page 5 it is unclear whether the authors make claims about CP as a global phenomenon or not. The info needs to be tied more specifically to the geopolitical location. What is missing is background information about the racialised and historically complex two-tier education system. Classrooms are very large and on the whole teachers not as qualified as in many other places. Therefore, this article could unfairly maybe give a very negative impression of a terribly underpaid and under supported workforce. Teachers in SA work under very difficult conditions and this article should include more information about teachers’ conditions so it doesn’t run the risk of being accused of a deficit approach, especially as teachers’ voice in the study is absent. It is only learners’ perspectives.

 We have revised the introduction to include background information about the racialized and historically complex two-tier education system in South Africa in line 34 - 38: 

Corporal punishment has been an integral part of schooling for most teachers and learners in twentieth century South African schools, characterized by a legacy of authoritarian education practices under Bantu education and a belief that CP is necessary for orderly education (7). The ending of apartheid and the establishment of a human rights culture in the 1990s laid the foundation for legislation aimed at ending use of CP in schools in South Africa (7). 

Line 40 – 44:

South Africa is a highly inequitable society and this inequality is reflected in the historically complex two-tier education system namely, the middle class, private schools and the public schools (5). The middle class, formerly white schools no longer use corporal punishment as a discipline method (7). However, in public schools, use of corporal punishment is still common practice (8, 9)

We agree with the reviewer that teachers, particularly in public schools, work under difficult conditions and this background information has now been added in line 44 – 47:

In most public schools, classrooms are overcrowded and under-resourced, and teachers are often under-qualified and overworked (10, 11). Teachers feel disempowered and ill-equipped with viable alternative discipline methods to maintain a safe and secure environment to facilitate learning (8)

4. Line 90: This is a generalisation not supported by evidence. Education departments in SA Universities seem to be doing their upmost to teach their students alternative discipline strategies, but the phenomenon is more complex than that. Also Provincial education departments all have initiatives to support teachers in using different discipline methods, but the particular image of child and childhood and child/adult relations tend to be hierarchical in SA and lack the equality needed for respectful non-violent relations. This is typical of patriarchal societies. The violence against children in SA is intricately connected to the violence against women.

 We have revised the paragraph. It now reads as follows in line 109 - 112:

While use of CP is illegal in South Africa, there has been limited concerted effort to enforce the law, ensuring that those who continue to use CP are convicted of an offense, and inadequate training of teachers on alternative methods of classroom management and discipline (30, 31)

More can be done to enforce implementation of legislation and holding accountable those who illegally continue to use CP despite legislation prohibiting its use. Gershoff points that: legal bans are not sufficient to completely eliminate school corporal punishment. Behavior change by teachers and school administrators is what is needed. Behaviour change will require consistent education about the harms of corporal punishment and about alternative, positive forms of discipline. While there might be pockets of schools where teachers are exposed to initiatives aimed to use positive discipline methods, not all schools have been provided with such, and the support has not been consistent. We have revised the conclusion, line 453 – 455:

There is an urgent need to break this cycle by enforcing the law, and holding accountable those who continue to use CP despite legislation prohibiting its use

And line 459 – 464:

Use of positive disciplining strategies, developing democratic relationship and consciousness about image of a child will direct how parents and teachers relate with children, critical for raising responsible children, and to curb future perpetration of violence in society (55, 56) Effecting engrained beliefs, attitudes and behaviours supportive of corporal punishment amongst parents and teachers will require consistent education about the harms of corporal punishment and support for why it needs to be ended. 

5. What is completely missing is the relationship between the training of teachers, their ability to teach in ways that engages learners and motivates them. Teacher educators know that e.g. more active and embodied pedagogies can be used that prevent 'misbehaviour' from occurring. Large class sizes etc also really contribute to CP and not having classroom assistants.

 We agree with the reviewer that there are many factors that could explain use of corporal punishment amongst teachers including factors related to their training. It is likely that these factors are contributing to their ability/ inability to teach in ways that engages the learner and motivates them, so as to manage some of the risk factors associated with learner experience of CP. As suggested by the reviewer, a study which captures teacher’s perspectives on use of corporal punishment in schools will contribute to knowledge and respond to these questions. We have not included teachers’ perspectives on this paper, and plan to do so in future. 

6. The intervention focuses on positive discipline and psychological interventions, but not on the image of the child or the pedagogical dimensions. So in a sense, the hypothesis of the study makes sense in the light of the identity of the researchers. It would be helpful to include in future research someone from a very different educational background so base-line hypotheses can be enriched. We know from educational research that teachers tend to teach how they were taught and it is very difficult to break through that despite the many interventions in teacher education and by provincial departments. Therefore, the suggestion that the solution to the problem identified might be to offer training in positive discipline is nothing new and has been going on for some time now. It is a good idea to check out this literature. What would be worthwhile and unusual and new instead is to include an in-service course/intervention in childhood studies (to teach about the image of the child) and the building of democratic relationships through e.g. ‘shared authority’ models. These are about more sustainable and durable changes as they could affect deeply engrained beliefs about children as found wanting (by adults), rather than interventions that don’t change fundamental beliefs about people and their relations. However, this would require pedagogical skills and working with both groups at the same time: learners and their teachers.

 We thank the reviewer for this comment, which we have noted. We will ensure that in future we involve someone with education background to help enrich our work by incorporating a focus on the image of the child or the pedagogical dimensions.

We have incorporated the reviewer’s suggestions on strategies to address corporal punishment in schools, line 459 – 464:

Use of positive disciplining strategies, developing democratic relationship and consciousness about image of a child will direct how parents and teachers relate with children, critical for raising responsible children, and to curb future perpetration of violence in society (55, 56) Effecting engrained beliefs, attitudes and behaviours supportive of corporal punishment amongst parents and teachers will require consistent education about the harms of corporal punishment and support for why it needs to be ended. 

In short, this is a very informative article, well written and researched and helpful in that it confirms much of what we already know about CP, and indeed helpful for funding purposes, but it misses the important geopolitical educational dimension (which still can be added) and references to pedagogies which hopefully can be taken into account in further studies. I hope the above comments are helpful, in order to add further reflections to the current paper, but especially I hope they are useful for future research purposes. We have revised the paper and included recommendations on teaching practices that are critical in efforts to end use of corporal punishment in schools, line 459 – 464:

Use of positive disciplining strategies, developing democratic relationship and consciousness about image of a child will direct how parents and teachers relate with children, critical for raising responsible children, and to curb future perpetration of violence in society (55, 56) Effecting engrained beliefs, attitudes and behaviours supportive of corporal punishment amongst parents and teachers will require consistent education about the harms of corporal punishment and support for why it needs to be ended. 

Reviewer 3 

This study examines the prevalence and factors associated with corporal punishment in public schools in South Africa using baseline data from a larger RCT study.

This is a highly competent and well published team who have completed most of the seminal work on interpersonal violence in South Africa and in fact globally

The methodology of the larger randomized controlled trial is excellent and it appears to be a well conducted study Thank you very much for your time reviewing our manuscript and for the valuable feedback.

I have one analytic concern. The authors merge two questions ‘In the past 6 months were you ever beaten by a teacher?’ and “A teacher might beat you or physically punish you at our school”. These are two entirely different questions and simply because most people answer yes to the first have answered yes to the second (of course they have) is no reason to merge them. This must be corrected in the revised version.

 Thank you for highlighting this point. We have corrected this. We have redefined the outcome using only 1 item (In the past 6 months were you ever beaten by a teacher?)

The use of the term ‘delinquent’ is highly problematic, certainly not justified by the data collected in the paper and should be deleted. To describe a learner as delinquent based on 5 questions is highly problematic. We have replaced the term delinquent with misbehaviour throughout the manuscript

A more general concern has to do with what these findings add to the literature. Corporal punishment is associated with learner behaviour, home environment and school environment (i.e. everything). It does feel a little like squeezing another paper out of the dataset. While the paper adds little to what is already known, the method and execution of the study are of a high standard and the findings may be of some interest to education authorities in South Africa. The revisions based on the comments from the reviewers have improved the paper. 

While a lot has been written about corporal punishment in schools, very few published papers discuss the prevalence and use SEM to describe the interrelationships between factors associated with learner experience of CP in schools in South Africa. Like in other countries, South African schools are contextually unique, and deserving to be studied so we can have a better understanding of factors associated with use of CP in schools in South Africa. Given that CP is prohibited, yet continues to be used says that there is a problem, which needs to be engaged with, and recommended made on how it can be addressed. 

As such, this paper contributes to literature on the prevalence and factors associated with learners experiences of corporal punishment in schools in South Africa, and makes recommendation on what can be done to curb the use of corporal punishment in schools.

References 

1. Republic of South Africa. South African Schools Act No 84 of 1996. In: Office Ps, editor. Pretoria, South Africa: Government Printers; 1996. p. 1 - 50.

2. Gershoff ET. School corporal punishment in global perspective: prevalence, outcomes, and efforts at intervention. Psychology, health & medicine. 2017;22(sup1):224-39.

3. Heekes S-L, Kruger CB, Lester SN, Ward CL. A systematic review of corporal punishment in schools: Global prevalence and correlates. Trauma, Violence, & Abuse. 2020:1524838020925787.

4. Munir A, Hussain B. Implications of Corporal Punishment on the Child's Mental Health in Peshawar, Pakistan. Pakistan Journal of Criminology. 2019;11(1).

5. Spaull N. Poverty & privilege: Primary school inequality in South Africa. International Journal of Educational Development. 2013;33(5):436-47.

6. Staeheli LA, Hammett D. ‘For the future of the nation’: Citizenship, nation, and education in South Africa. Political Geography. 2013;32:32-41.

7. Morrell R. Corporal punishment in South African schools: A neglected explanation for its existence. South African Journal of Education. 2001;21(4):292-9.

8. Maphosa C, Shumba A. Educators' disciplinary capabilities after the banning of corporal punishment in South African schools. South African Journal of Education. 2010;30(3):0-.

9. Breen A, Daniels K, Tomlinson M. Children's experiences of corporal punishment: a qualitative study in an urban township of South Africa. Child abuse & neglect. 2015;48:131-9.

10. Phurutse MC. Factors affecting teaching and learning in South African public schools: HSRC press; 2005.

11. Mji A, Makgato M. Factors associated with high school learners' poor performance: a spotlight on mathematics and physical science. South African journal of education. 2006;26(2):253-66.

12. Marumo M, Zulu C. Teachers’ and learners’ perceptions of alternatives to corporal punishment: A human rights perspective. In: Zulu C, Oosthuizen I, Wolhuter C, editors. A scholarly inquiry into disciplinary practices in educational institutions. 2. Cape Town, South Africa: AOSIS; 2019. p. 45 - 66.

---

## [Decision Letter · Decision Letter 1]

28 Jun 2021

Prevalence and factors associated with experience of corporal punishment in public schools in South Africa

PONE-D-20-20199R1

Dear Dr. Mahlangu,

We’re pleased to inform you that your manuscript has been judged scientifically suitable for publication and will be formally accepted for publication once it meets all outstanding technical requirements.

Kind regards,

Thach Duc Tran, M.Sc., Ph.D.

Academic Editor

PLOS ONE

Additional Editor Comments (optional):

Reviewers' comments:

Reviewer's Responses to Questions

**Comments to the Author**

1. If the authors have adequately addressed your comments raised in a previous round of review and you feel that this manuscript is now acceptable for publication, you may indicate that here to bypass the “Comments to the Author” section, enter your conflict of interest statement in the “Confidential to Editor” section, and submit your "Accept" recommendation.

Reviewer #1: All comments have been addressed

Reviewer #2: All comments have been addressed

2. Is the manuscript technically sound, and do the data support the conclusions?

Reviewer #1: Yes

Reviewer #2: Yes

3. Has the statistical analysis been performed appropriately and rigorously? 

Reviewer #1: Yes

Reviewer #2: I Don't Know

4. Have the authors made all data underlying the findings in their manuscript fully available?

Reviewer #1: Yes

Reviewer #2: Yes

5. Is the manuscript presented in an intelligible fashion and written in standard English?

Reviewer #1: Yes

Reviewer #2: Yes

6. Review Comments to the Author

Reviewer #1: COMMENTS: Most of the comments made on earlier draft {by me and hopefully by other respected reviewers also) were/are attended adequately. The manuscript is improved a lot. May be accepted if found clinically relevant.

Reviewer #2: Thank you for addressing my concerns quite meticulously. One of my key concerns does remain though and that is the narrow way in which CP has been defined in the study. Of course that can no longer be retified as it has been asumed from the outset and informed tha data collection. Corporal punishment is broader than 'being beaten'. Research has thrown up how much pain and physical discomfort is caused without beating a child. This can be by letting children stand for hours in a hot sun with their hands above their heads and without water or let them stand in a particular position against the wall for a long time so their legs will start to hurt. There are many ways in which children are hurt daily without being beaten. That is also corporal punishment. See eg the following link for the compexity in defining CP. This needs to be better explored and addressed in future studies. https://www.un.org/development/desa/family/wp-content/uploads/sites/23/2020/06/Freer_Expert-Group-Paper_Corporal-Punishment-Physical-Abuse_June2020.pdf

7. PLOS authors have the option to publish the peer review history of their article (what does this mean?). If published, this will include your full peer review and any attached files.

Reviewer #1: **Yes: **Dr. Sanjeev Sarmukaddam

Reviewer #2: **Yes: **Professor Karin Murris

---

## [Editor Report · Acceptance letter]

4 Aug 2021

PONE-D-20-20199R1 

Prevalence and factors associated with experience of corporal punishment in public schools in South Africa. 

Dear Dr. Mahlangu:

I'm pleased to inform you that your manuscript has been deemed suitable for publication in PLOS ONE. Congratulations! Your manuscript is now with our production department. 

Kind regards, 

on behalf of

Dr. Thach Duc Tran 

Academic Editor

PLOS ONE